# Glial Markers of Suicidal Behavior in the Human Brain—A Systematic Review of Postmortem Studies

**DOI:** 10.3390/ijms25115750

**Published:** 2024-05-25

**Authors:** Mana Yamamoto, Mai Sakai, Zhiqian Yu, Miharu Nakanishi, Hatsumi Yoshii

**Affiliations:** 1Department of Psychiatric Nursing, Graduate School of Medicine, Tohoku University, 2-1 Seiryo-machi, Aoba-ku, Sendai 980-8575, Japan; 2Department of Psychiatry, Graduate School of Medicine, Tohoku University, 1-1 Seiryo-machi, Aoba-ku, Sendai 980-8573, Japan

**Keywords:** astrocyte, microglia, oligodendrocyte, suicidal behavior, postmortem

## Abstract

Suicide is a major public health priority, and its molecular mechanisms appear to be related to glial abnormalities and specific transcriptional changes. This study aimed to identify and synthesize evidence of the relationship between glial dysfunction and suicidal behavior to understand the neurobiology of suicide. As of 26 January 2024, 46 articles that met the inclusion criteria were identified by searching PubMed and ISI Web of Science. Most postmortem studies, including 30 brain regions, have determined no density or number of total Nissl-glial cell changes in suicidal patients with major psychiatric disorders. There were 17 astrocytic, 14 microglial, and 9 oligodendroglial studies using specific markers of each glial cell and further on their specific gene expression. Those studies suggest that astrocytic and oligodendroglial cells lost but activated microglia in suicides with affective disorder, bipolar disorders, major depression disorders, or schizophrenia in comparison with non-suicided patients and non-psychiatric controls. Although the data from previous studies remain complex and cannot fully explain the effects of glial cell dysfunction related to suicidal behaviors, they provide risk directions potentially leading to suicide prevention.

## 1. Introduction

Suicide is the cause of more than 700,000 deaths every year worldwide, resulting in complicated grief and dire economic crises among bereaved families. Given the broad range of the impact, suicide prevention constitutes a major health priority [1]. The epidemiology of suicide is multifactorial, involving a series of psychosocial factors, economic losses, and biological processes, most of which are related to psychiatric diseases (60–98%) [2]. Indeed, several epidemiological studies have identified the direct risk of mental illnesses as risk factors for suicide, such as major depressive disorder (MDD), unipolar and bipolar depression (UD/BD), schizophrenia (SCZ), and affective disorders (AD) [3]. Notably, mental distress is generally characterized by genetic and biological dysfunction involving brain circuits and network abnormalities [4], and the potential genetic and biological risks are focused on the elucidation of pathogenesis and biomarker identification for suicidal behaviors that infer suicide prevention [5].

In the central nervous system (CNS), non-neuronal glial cells, including astrocytes, microglia, and oligodendrocytes, are major satellite cells with specialized functions that maintain brain homeostasis, defense against invasion, and stress [6,7]. Several studies have reported an association between glial cells and suicidal behaviors based on postmortem brain gene expression [8], morphological changes [9], cytokines [10], and neurotrophic factors [11]. Astrocytes are the most abundant glial cells in the brain and regulate synaptic transmission and plasticity [7]; they mediate myelinated axonal excitability and conduction velocity [12], synchronization with the activity of adjacent neurons [13], and neuronal communication between brain regions [14]. They are also known to contact projections to the vascular basement membrane of the brain to help maintain the closure of the blood–brain barrier (BBB) and form the spinal fluid–brain barrier on the surface of the brain [15]. In addition, astrocytes synthesize and release diverse polypeptides, including brain-derived neurotrophic factors, which contribute to the maintenance of diverse brain activities such as dendritic neuron growth, axonal outgrowth, and neuronal circuit repair, and are associated with mood disorders [16,17]. Microglia account for approximately 10% of cells and are the most abundant resident mononuclear phagocytes in the CNS [18], playing roles in brain development [19] and cell–cell communication maintenance [20]. Microglial markers, including chemokine receptors such as C-X3-C motif chemokine receptor 1 (CX3CR1), may affect blood volume in the human brain [21]. Furthermore, we reported that the microglial response to acute and chronic stress influences cytokine production, such as interferon-alpha/beta/gamma (IFNα/β/γ), tumor necrosis factor (TNF-α), interleukin-1beta (IL-1β), IL-10, and transforming growth factor-beta 1 (TGF-β), and neurotransmitters [22] and behaviors in animal models [23,24,25,26]. Oligodendrocytes are myelinating cells of the CNS that account for about 5–10% of the total glia [27]. They are assumed to be crucial in forming the complex neural circuits required for cognitive function [28] by producing myelin sheaths and the rapid and efficient conduction of electrical impulses along axons [28]. Recently, it has been suggested that trait changes and dysfunction associated with oligodendrocytes may be related to psychiatric disorders such as MDD [29].

Glial cell dysfunction has been implicated in the pathophysiology of MDD [30], SCZ [31], and suicidal behavior [32]. However, there is currently no direct quantitative comparison of evidence across individual glial biomarkers for suicidal behavior. We reviewed all available evidence of glial-related alterations in postmortem suicidal brains to clarify the crucial role of glial pathology in suicidal behaviors.

## 2. Materials and Methods

Following the guidelines of the Preferred Reporting Items for Systematic reviews and Meta-Analyses (PRISMA) statement [33], we conducted this review using prospective protocol. The review protocol is registered with PROSPERO (registered number: CRD42024537167).

### 2.1. Literature Search Strategy

This systematic review of postmortem brain studies was conducted to determine the association between glial cells and suicidal behavior. Studies were identified by searching the PubMed and ISI Web of Science databases for peer-reviewed journal articles published until January 2024. The above databases were searched using the following keyword combinations: (“suicid*”) AND (“glia” OR “microglia” OR “astrocyt*” OR “oligodendrocyt*”).

Review of patients, interventions, comparators, outcome measures, and study designs (PICO framework):

Patient, Population, or Problem: human who died from intentional self-harm.

Intervention: not applicable.

Comparison: human who died from sudden death or natural course.

Outcome: glial biomarkers,

We asked, “Do glial markers exist in postmortem brain samples from individuals who committed suicide?

### 2.2. Eligibility Criteria

We included original peer-reviewed articles that met the following criteria: (i) written in English; (ii) analyzed the association between glial density, glial-related genes, and suicidal behavior; and (iii) were case–control studies comparing healthy controls whose cause of death was sudden or natural death with those who died by suicide. We excluded duplicate papers, in vitro studies, review articles, animal studies, studies with unknown causes of death, studies that focused on self-harm, and studies that did not report glial genes in postmortem brain samples.

### 2.3. Screening

In the first screening, 590 articles were included (262 from PubMed and 328 from Web of Science). A total of 281 duplicates were excluded using reference management software (Endnote for mac, version X9; Clarivate Analytics, London, UK). The retrieved studies were screened based on their titles and abstracts, and 211 articles were omitted based on the exclusion criteria. At the end of the second screening, 46 articles were included after 52 articles were discarded (Figure 1).

### 2.4. Article Selection

The titles and abstracts were screened for eligibility by the authors (M.Y., M.S., and Z.Y.). Simultaneous assessment by the three primary authors resolved the controversies regarding these studies. After agreement among the three authors, valid references based on the selection criteria were selected for final inclusion, and full-text PDFs were obtained and analyzed. The two authors, M.S. and Z.Y., resolved any issues whenever a consensus could not be reached between the first three authors.

### 2.5. Data Extraction

Data were methodically extracted from extensively reviewed journal articles. The variables extracted for our review included gene symbols, expression type, brain region, and glial markers, as well as the manner in which the glial marker profile was associated with suicide.

## 3. Results

### 3.1. Glial-Related Alterations in Postmortem Suicidal Brain

#### 3.1.1. Glial Cell Populations

A total of 27 studies characterized the number and densities of glial cells and their changes in transcription and methylation during suicides. Table 1 shows two studies focused on the history of alcohol dependence and three studies on early life adversity (child abuse), in which 25 studies compared psychiatric suicide patients (AD, BD, MDD, SCZ, and UD) with non-suicide patients with or without psychiatric disorders. Among these studies, seven reports of glia distinct from neurons were identified by Nissl (thionine) and NeuN staining. Furthermore, five studies determined density alterations in proteins of glial fibrillary acidic protein (GFAP) or Vimentin-immunoreactive (IR) astrocytes. Nine studies identified the density and number of microglia using specific protein antibodies, including human leukocyte antigen-DR isotype (HLA-DR), ionized calcium-binding adapter molecule 1 (IBA1), and N-methyl-D-aspartate (NMDA) glutamate receptor agonist quinolinic acid (QUIN)-IR. Five studies focused on oligodendrocytes by using the protein antibodies of transcription factor 2 (OLIG2), myelin basic protein (MBP), or platelet-derived growth factor receptor alpha (PDGFRA) positive oligodendrocyte-lineage (OL) cells and RNA-Seq analysis in the suicides (Figure 2 and Table 1).

According to Nissl and NeuN staining, there was no significant difference in numerical density in the orbitofrontal cortex (OFC) [34], anterior cingulate cortex (ACC) [35], primary auditory cortex (PAC) [36], auditory association cortex (AAC) [36], or dorsomedial prefrontal cortex (DLPFC) [37] between suicide patients and matched controls. Furthermore, Nissl staining showed no significant changes in glial volumes in the dentate gyrus (DG) and hilus region of the cornu ammonis (CA) of the hippocampus (HIPP) [38,39] in depressed suicide victims. Although not in our review searched targets, glial densities also did not change in the basolateral amygdala of depressed suicides [40]. However, among alcohol-dependent depressed suicide decedents, there were higher glial cell densities in the BA24 (ACC) than in suicide decedents who were not alcohol-dependent or controls [35]. A small sample study (n = 4) showed that the number of glial cells was altered in the anterior DG of HIPP, with decreased glial cells in unmedicated suicides compared to controls [41]. Early life adversity (<15 years of age) exposure was associated with a trend of more glia in the whole DG in depressive suicide patients and matched controls [38].

In astrocyte research, Golgi-stained cortical astrocytic morphology showed significantly larger cell bodies and numbers of nodes, average number of branches, total branch length, and total number of spines as well as more ramified processes in the ACC of depressed suicides, suggesting a neuroinflammatory theory of depression [42]. Similarly, there was no difference in the density of GFAP-positive astrocyte protein in any of the hippocampal regions between patients with and without depressive suicide, whereas the fraction of GFAP immunoreactivity in the CA2/3 area was inversely correlated with the duration of depression in suicide victims [43]. However, O’Leary et al. determined that the cerebral protein of GFAP and Vimentin-IR astrocyte densities were reduced in depressed suicide patients [44]. Furthermore, in a postmortem study of the locus coeruleus (LC) of MDD, most of the decedents who had MDD and died by suicide showed a significantly lower density of GFAP-IR astrocytes than matched controls, but not Nissl-stained oligodendrocytes [45]. Ernst et al. could not detect a difference in the expression level of *GFAP* astrocyte between controls (including alcohol dependence but without BD or MDD) and depressed suicides but significantly low levels of a truncated variant of tropomyosin-related kinase B (*TrkB.T1*) (highly expressed in astrocytes) [46]. Their further study confirmed the hypermethylated 3′UTR region of *TrkB.T1* in the DLPFC and the frontal eye field (FEF) of suicide completers that were selected with low *TrkB.T1* expression compared with psychiatrically normal controls [47]. An aged suicide study found that patients with BD exhibit increased transcription but decreased protein levels of GFAP in the ACC, whereas patients with MDD do not [48].

Steiner et al. used ANOVA to determine significant microgliosis (HLA-DR protein) in suicide patients in the DLPFC, ACC, and mediodorsal thalamus (MD), with a similar trend in the HIPP (*p* = 0.057) but without a post hoc test to determine whether specific groups are responsible for the significant differences [49,50]. Furthermore, the protein of IBA1-immunoprecipitation (IP) microglial cell density in the anterior midcingulate cortex (AmCC) was significantly increased in suicidal BD subjects compared to BD non-suicidal controls, but there were no significant changes in microglial density between SCZ suicides and non-suicides [51]. Moreover, depressive suicides showed significantly increased protein of QUIN-IP microglial density in the AmCC and subgenual ACC (sACC) but not in the pregenual ACC (pACC), compared to controls. QUIN-IP microglial density was increased in suicides with MDD [52], whereas it was reduced in the right hippocampal CA1 of UD and BD suicides [53]. Furthermore, in the protein of major histocompatibility complex class II (MHC-II; HLA-DP, DQ, DR), the microglial density and activation are significantly elevated in the HIPP of patients with suicidal BD, whereas non-suicidal BD patients are similar to controls [54]. In contrast, a quantitative protein analysis of HLA-DR microglial density in the dorsal raphe nucleus (DRN) revealed no significant differences in microglial density between suicidal and non-suicidal patients from both diagnostic groups of psychiatric disorders (MDD, BD, or SCZ) and controls [55]. The total density of each type (reactive/primed/amoeboid) of IBA1 protein IR microglia also did not differ between depressed suicides and controls in the dorsal anterior cingulate cortex (dACC) region [56], dorsal or ventral prefrontal white matter (including SCZ and AD) [57], or the expression of microglial QUIN in the left hippocampal CA1 and right CA2/3 [53].

A previous study confirmed that MBP protein immunoreactivity was decreased in individuals with SCZ and depression who died by suicide compared to psychiatrically normal controls [58]. Furthermore, Tanti et al. reported the decreased density of oligodendrocytes in depressed suicide victims with a history of child abuse that showed oligodendroglial abnormalities, significantly increased numbers of mature myelinating oligodendrocytes, accompanied by decreased numbers of more immature oligodendrocyte-lineage cells in the ventromedial prefrontal cortex (VMPFC) by using the proteins OLIG2, MBP, PDGFRA, adenomatous polyposis coli protein (APC), and neurite outgrowth inhibitor (NogoA) maturational stage-specific markers [59].

Laser capture showed that the relative telomere lengths of oligodendrocytes in both the white matter of the LC and occipital cortex (OCC) were significantly shorter in patients with suicidal MDD than in psychiatrically normal controls [60]. A recent RNA-Seq analysis identified that the transcription of oligodendrocyte cell populations in the temporal cortex (TC) was significantly lower in a depressed suicide group than in psychiatrically normal controls [61]. In addition, a differentially methylated region analysis of the OFC and ventral anterior cingulate cortex (vACC) from the database indicated significant changes in the cell type proportions of astrocytes, microglia, and oligodendrocytes [62].

**Table 1 ijms-25-05750-t001:** Changes in densities and numbers of glial cells in suicides.

No.	Cell Type	Region	Changes	Diagnoses	Comparison
1	Glia (Nissl)	OFC	→ [34]	MDD (Ad)	Ad suicides vs. Ad non-suicides
2	Glia (Nissl)	ACC	→ vs. non-suicides or ↑ vs. non-Ad suicides [35]	MDD (Ad)	Ad suicides vs. non-Ad suicides Ad suicides vs. non-suicides
3	Glia (Nissl)	ACC and DLPFC	→ [37]	MDD (ELA) and SCZ (ELA)	ELA suicides vs. ELA non-suicides
4	Glia (Nissl)	AAC and PAC	→ [36]	MDD, SCZ	MDD and SCZ suicides vs. H
5	Glia (Nissl)	CA and DG	→ [38]	MDD (ELA)	MDD (ELA) suicides vs. ELA non-suicides
6	Glia (Nissl)	CA and DG	→ [39]	MDD, SCZ	MDD and SCZ suicides vs. H
7	Glia (Nissl)	anterior DG	↓ [41]	MDD (As)	MDD (unmedicated) suicides vs. H
8	Astrocyte (Golgi)	ACC	↑ [42]	MDD	MDD suicides vs. H
9	Astrocyte (GFAP)	CA	→ [43]	MDD	MDD suicides vs. H
10	Astrocyte (GFAP and Vimentin)	DCN, DLPFC, and MD	↓ [44]	MDD	MDD suicides vs. H
11	Astrocyte (GFAP), synemin-α, synemin-β, Vimentin, nestin)	ACC and DLPFC	↑↓ [48]	BD and MDD	BD and MDD suicides vs. H
12	Astrocyte (GFAP)	LC	↓ [45]	MDD	MDD suicides vs. H
13	Astrocyte (*GFAP*, *TrkB.T1*, Trkb.T1)	DLPFC	→ [46]	BD and MDD	MDD suicides vs. H
14	Microglia (HLA-DR)	ACC, DLPFC, and MD	↑ [50] ^$^	AD, SCZ	AD and SCZ suicides vs. AD and SCZ non-suicides
15	Microglia (HLA-DR)	ACC and MD	↑ [49] ^#^	SCZ	SCZ suicides vs. SCZ non-suicides
16	Microglia (HLA-DR)	DRN	→ [55]	BD, MDD, and SCZ	BD, MDD, and SCZ suicides vs. BD, MDD, and SCZ non-suicides vs. H
17	Microglia (MHC II, P2RY12)	HIPP	↑ [54]	BD	BD suicides vs. BD non-suicides
18	Microglia (IBA1)	AmCC	↑ [51]	BD, SCZ	BD and SCZ suicides vs. H
19	Microglia (IBA1)	dACC	→ [56]	UD	UD suicides vs. H
20	Microglia (IBA1)	DLPFC and VLPFC	→ [57]	AD, SCZ	AD, SZ, and H suicides vs. AD, SZ, and H non-suicides
21	Microglia (QUIN)	sACC and AmCC	↑ [52]	BD, MDD	BD and MDD suicides vs. H
22	Microglia (QUIN)	right CA1	↓ [53]	BD, UD	BD and UD suicides vs. H
23	Oligodendrocyte (OLIG2, PDGFRA, APC, and NogoA)	VMPFC	↑↓ [59]	MDD (ELA)	MDD (ELA) suicides vs. MDD suicides MDD (ELA) suicides vs. H
24	Oligodendrocyte (MBP)	aPFC	↓ [58]	MDD and SCZ	MDD, SCZ suicides vs. H
25	Oligodendrocyte population	TC	↓ [61]	MDD	MDD suicides vs. H
26	Oligodendrocyte (telomere lengths)	OCC and LC	↓ [60]	MDD	MDD suicides vs. H
27	Methylated regions proportions	OFC and vACC	Changed [62]	MDD	MDD suicides vs. H

Ad, alcohol dependence. As, antidepressants. AD, affective (mood) disorders. ELA, early-life adversity/child abuse. APC, adenomatous polyposis coli protein. BD, bipolar disorder. MDD, major depressive disorder. SCZ, schizophrenia. UD, unipolar depression. GFAP, glial fibrillary acidic protein. HLA-DR, human leukocyte antigen-DR isotype. OLIG2, oligodendrocyte transcription factor 2. PDGFRA, platelet-derived growth factor receptor alpha. QUIN, N-methyl-D-aspartate glutamate receptor agonist quinolinic acid. Myelin basic protein, MBP. Truncated splice variant of tropomyosin-related kinase B (TrkB.T1). OFC, orbitofrontal cortex. ACC, anterior cingulate cortex. PAC, primary auditory cortex. AAC, auditory association cortex. DG, dentate gyrus. CA, cornu ammonis. BLA, basolateral amygdala. LC, locus coeruleus. MD, mediodorsal thalamus. DLPFC, dorsomedial prefrontal cortex. DRN, dorsal raphe nucleus. sACC, subgenual anterior cingulate cortex. AmCC, anterior midcingulate cortex. dACC, dorsal anterior cingulate cortex. pACC, pregenual anterior cingulate cortex. aPFC, anterior prefrontal cortex. VLPFC, ventrolateral prefrontal cortex. VMPFC, ventromedial prefrontal cortex. HIPP, hippocampus. OCC, occipital cortex. TC, temporal cortex. DCN, dorsal caudate nucleus. vACC, ventral anterior cingulate cortex. H, neuropsychiatric healthy non-suicide control. ^$^, without post hoc statistical significance. ^#^, without statistical analysis. ↑, increased. ↓, decreased. →, unchanged.

**Figure 2 ijms-25-05750-f002:**
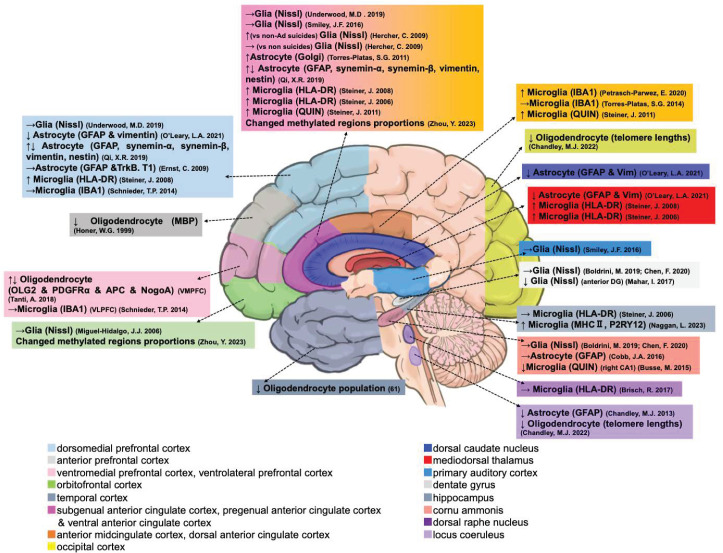
Postmortem evidence of glial abnormalities in different brain regions. ↑, increased. ↓, decreased. →, unchanged [34,35,36,37,38,39,41,42,43,44,45,46,48,49,50,51,52,53,54,55,56,57,58,59,60,61,62].

#### 3.1.2. Astrocytes

Astrocytes are the most functionally diverse glial cells in the CNS, including modulation of BBB and their permeability and provision of nutrients to the nervous tissue [15]. There were 17 studies associated with alterations in the transcription, protein content, and methylation of astrocyte-related suicides in the current review (Figure 3 and Table 2). In suicide decedents, real-time polymerase chain reaction (PCR) determined *GFAP*, a major marker for astrocytes, and its transcription were decreased in the DLPFC [63], MD, caudate nucleus (CN) [64], and LC [45] but not in the primary motor cortex (PMC), primary visual cortex (PVC), or cerebellar cortex (Cb) [64]. In contrast, Zhang et al. did not confirm changes in *GFAP* mRNA levels in the DLPFC and ACC between suicide and non-suicide patients with SCZ and their matched controls [65]. Ernst et al. could not detect a difference in *GFAP* mRNA between depressed suicides and controls in the FEF and DLPFC but found low expression of *TrkB.T1* [46].

Furthermore, immunoblotting and -staining have shown decreased levels of GFAP protein in the DLPFC [44,63], MD [44,64], CN [44,64], HIPP [43], and LC [45], along with decreased Vimentin protein in the DLPFC, MD, and CN [44], but not in the PMC, PVC, and Cb [64]. Moreover, Schlicht et al. identified prefrontal cortical GFAP (sp4005) protein levels only in the suicides, suggesting that a higher phosphorylation state especially in the GFAP might have an impact on the pathophysiology of suicidal behavior [66]. Additionally, there was no difference in the expression levels of *GFAP* mRNA between the controls (including alcohol dependence but without BD or MDD) and the depressed suicide completers with low levels of *TrkB.T1* mRNA and protein expression [46]. As described previously, older suicide patients with BD exhibit increased transcription and decreased GFAP protein levels in the ACC [48].

Previous studies also reported decreased transcription levels of several astrocytic markers in suicide decedents, including aldehyde dehydrogenase 1 family member L1 (*ALDH1L1*) [63,65], SRY-Box Transcription Factor 9 (*SOX9*) [63,67], glutamate-ammonia ligase (*GLUL*) [63,68], S100 calcium-binding protein B (*S100B*) [68], and solute carrier family 1 member 3 (*SLC1A3*) [63] in the DLPFC, and both *SLC1A3* and solute carrier family 1 member 2 (*SLC1A2*) in the LC [45]. Furthermore, literature not in our searched target indicated that astroglia located *SLC1A3* and *GLUL* transcripts decreased in the DLPFC of suicidal MMD patients [69]. The other two primary components of intercellular gap junction channels that are most abundant in astrocytes are gap junction protein alpha 1 (*GJA1*) and beta 6 (*GJB6*) [70], and their transcriptions are decreased in the ACC [71], DLPFC [63,67], MD, PMC, PVC, CN, but decreased *GJA1* and increased *GJB6* in the Cb [72]. Moreover, Miguel-Hidalgo et al. determined decreased protein levels of intercellular adhesion molecule 1 (ICAM1) [73] in the OFC of suicide victims but not GJA1 protein [74]. García-Gutiérrez et al. identified cannabinoid receptor 2 (CNR2) and the non-cannabinoid receptor G Protein-Coupled Receptor 55 (GPR55) in the DLPFC postmortem at both the mRNA and protein levels, which decreased CNR2 and *GPR55* mRNA levels [75]. In contrast, the protein levels of CNR2 and CNR2-GPR55 receptor heteromers were increased in suicide decedents, but this was not the case for GPR55 protein [75]. In contrast, a previous study did not confirm the changes in the transcription of *ALDH1L1* and *S100B* in depressed suicides compared to depressed patients and matched controls in the ACC [76]. In addition, a postmortem study of glial gene expression in patients with and without SCZ confirmed decreased transcription of glutamine synthetase (*GS; GLUL*) in DLPFC suicides with SCZ [65].

DNA methylation and chromatin modifications have also been confirmed in astrocyte-associated genes in depression-related suicide cases. For instance, Nagy et al. confirmed that methylation status was decreased in astrocytic DNA glutamate ionotropic receptor kainate type subunit 2 (*GRIK2*) and nebulette (*NEBL*) but increased in nectin cell adhesion molecule 3 (NECTIN3; *PVRL3*) and rhophilin-associated tail protein 1B (*ROPN1B*) in the DLPFC of depressed suicide patients [63]. Nagy et al. showed that decreased transcription of *GJB6* and *GJA1* are likely mediated by the repressive histone modification of H3K9me3 in the DLPFC [72]. The *TrkB.T1* gene transcript, which is highly expressed in astrocytes, is hypermethylated in depressed suicide patients in the DLPFC (BA8/9) compared to psychiatrically normal controls [47].

**Table 2 ijms-25-05750-t002:** Changes in densities and numbers of astrocytic cells in suicides.

Astrocyte	Type	ACC	FEF	DLPFC	OFC	MD	PMC	PVC	CN	HIPP	LC	Cb
Marker												
*GFAP*	mRNA	↑ [48] → [65]	→ [46]	↓ [63] → [46,65]	—	↓ [64]	→ [64]	→ [64]	↓ [64]	—	↓ [45]	→ [64]
GFAP	Protein	↓ [48]	—	↑ [66]	—	↓ [64]	→ [64]	→ [64]	↓ [64]	—	↓ [45]	→ [64]
	Stain	—	—	↓ [44,63]	—	↓ [44]	—	—	↓ [44]	↓ [43]	—	—
Vimentin	Stain	—	—	↓ [44]	—	↓ [44]	—	—	↓ [44]	—	—	—
*ALDH1L1*	mRNA	→ [76]	—	↓ [63,65] → [76]	—	—	—	—	—	—	—	—
*SOX9*	mRNA	—	—	↓ [63,67]	—	—	—	—	—	—	—	—
*GLUL*	mRNA	→ [65]	—	↓ [63,65] ↑↓ [68]	—	—	—	—	—	—	—	—
*S100B*	mRNA	→ [76]	—	↑↓ [68] → [76]	—	—	—	—	—	—	—	—
*GJA1 (CX43)*	mRNA	—	—	↓ [63,67]	—	↓ [72]	↓ [72]	↓ [72]	↓ [72]	—	—	↓ [72]
GJA1 (CX43)	Protein	—	—	—	→ [74]	—	—	—	—	—	—	—
*GJB6 (CX30)*	mRNA	↓ [71]	—	↓ [63,67]	—	↓ [72]	↓ [72]	↓ [72]	↓ [72]	—	—	↑ [72]
ICAM1	Stain	—	—	—	↓ [73]	—	—	—	—	—	—	—
*CNR2*	mRNA	—	—	↓ [75]	—	—	—	—	—	—	—	—
CNR2	Protein	—	—	↑ [75]	—	—	—	—	—	—	—	—
*GPR55*	mRNA	—	—	↓ [75]	—	—	—	—	—	—	—	—
GPR55	Protein	—	—	→ [75]	—	—	—	—	—	—	—	—
CNR2-GPR55	Protein	—	—	↑ [75]	—	—	—	—	—	—	—	—
CRYAB	Protein	—	—	↑ [66]	—	—	—	—	—	—	—	—
*TrkB.T1*	mRNA	—	↓ [46]	↓ [46]	—	—	—	—	—	—	—	—
TrkB.T1	Protein	—	↓ [46]	↓ [46]	—	—	—	—	—	—	—	—
Glutamate–glutamine cycle												
*SLC1A3 (EAAT1)*	mRNA	—	—	↓ [63]	—	—	—	—	—	—	↓ [45]	—
*SLC1A2 (EAAT2)*	mRNA	—	—	—	—	—	—	—	—	—	↓ [45]	—
Methylation												
GRIK2	DNA	—	—	↓ [63]	—	—	—	—	—	—	—	—
NEBL	DNA	—	—	↓ [63]	—	—	—	—	—	—	—	—
PVRL3	DNA	—	—	↑ [63]	—	—	—	—	—	—	—	—
ROPN1B	DNA	—	—	↑ [63]	—	—	—	—	—	—	—	—
*TrkB.T1*	DNA	—	—	↑ [47]	—	—	—	—	—	—	—	—
Epigenetic silencing												
H3K9me3	DNA	—	—	↑ [72]	—	—	—	—	—	—	—	—

GFAP, glial fibrillary acidic protein. ALDH1L1, aldehyde dehydrogenase 1 family member L1. SOX9, SRY-Box Transcription Factor 9. GLUL, glutamate-ammonia ligase. GJB6 (CX30), gap junction protein beta 6. GJA1 (CX43), gap junction protein alpha 1. S100B, S100 calcium-binding protein B. CNR2, cannabinoid receptor 2. GPR55, G protein-coupled Receptor 55. CNR2-GPR55, CNR2 and GPR55 heteroreceptor complexes. ICAM-1, intercellular adhesion molecule 1. CRYAB, crystallin Alpha B. SLC1A3, solute carrier family 1 member 3. SLC1A2, solute carrier family 1 member 2. SLC1A1, solute carrier family 1 member 1. SLC1A6, solute carrier family 1 member 6. SLC38A1, solute carrier family 38 member 1. SLC38A2, solute carrier family 38 member 2. SLC1A4, solute carrier family 1 member 4. SLC1A5, solute carrier family 1 member 5. GRIK2, glutamate ionotropic receptor kainate type subunit 2. NEBL, nebulette. PVRL3 (NECTIN3, CD113), nectin cell adhesion molecule 3. ROPN1B, rhophilin-associated tail protein 1B. TrkB.T1, tropomyosin-related kinase B. H3K9me3, trimethylation of lysine 9 on histone H3. FEF, frontal eye fields. DLPFC, dorsolateral prefrontal cortex. OFC, orbitofrontal cortex. MD, mediodorsal nucleus. PMC, premotor cortex. PVC, primary visual cortex. CN, caudate nucleus. HIPP, hippocampus. LC, locus coeruleus. Cb, cerebellum. ↑, increased; ↓, decreased; →, unchanged; — not applicable.

**Figure 3 ijms-25-05750-f003:**
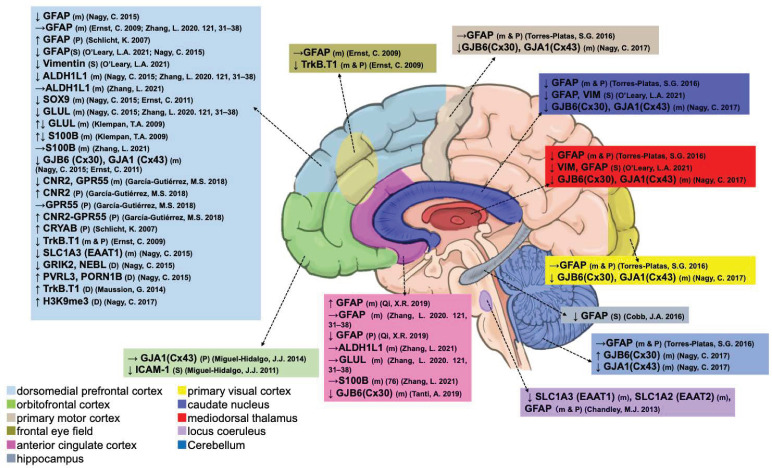
Postmortem evidence of astrocytic abnormalities in different brain regions. m, mRNA. P, protein. S, stain. D, DNA. ↑, increased. ↓, decreased. →, unchanged [43,44,45,46,47,48,63,64,65,66,67,68,71,72,73,74,75,76].

#### 3.1.3. Microglia

As the primary immune cell of the CNS [9], fourteen papers reported on the alterations in transcription, protein content, and methylation of microglia-related suicides (Figure 4 and Table 3). Zhang et al. confirmed that the transcription of three major microglial markers, *CX3CR1*, purinergic receptor P2Y12 (*P2RY12*), and triggering receptor expressed on myeloid cells 2 (*TREM2*) was increased in the ACC [65] and HIPP [54] of suicide patients with SCZ or BD compared with psychiatric non-suicidal individuals. Punzi et al. compared differentially expressed genes (DEGs) in the PFC between violent and non-violent suicide patients (BD, MD, or SCZ) and determined increased transcription of *P2RY12*, purinergic receptor P2Y13 (*P2RY13*), and G protein-coupled receptor 34 (*GPR34*) (a paralog of P2RY14), which are enriched in the purinergic signaling of microglia [77]. Naggan et al. also confirmed that the protein levels of microglial lymphocyte activation gene 3 (LAG3) and MHC II, which are involved in the depressive and antidepressant effects of electroconvulsive therapy, were significantly increased and positively correlated with HIPP in patients with suicidal BD [54]. Elevated transcription of *P2RY12*, but not *TREM2*, in the DLPFC along with increased *CD68* was also confirmed in suicidal patients with BD compared with non-suicidal patients [78]. In contrast, Zhang et al. only confirmed elevated *CD68* transcription in the DLPFC but not in the ACC in depressed suicide patients compared with psychiatrically normal controls. Furthermore, neither transcriptions of microglial protein tyrosine phosphatase receptor type C (*PTPRC*), *CX3CR1*, *P2RY12*, *HLA-DRA*, *TREM2*, integrin subunit alpha M (*ITGAM; CD11B*), nor translocator protein (*TSPO*) were elevated in non-suicidal patients with MDD [76]. However, in an imaging study using [^11^C] (R)-PK11195 positron emission tomography, TSPO availability predominantly in microglia was significantly increased in patients with MDD with suicidal thoughts compared with those without suicidal thoughts in the ACC and insula but not DLPFC [79].

Based on HLA-DA immunostaining, Steiner et al. revealed a single case of SCZ with higher microglial density in the ACC and MD who had committed suicide [49]. Their study also indicated that suicide patients with psychiatric disorders exhibited significant microgliosis (HLA-DR) in the ACC, DLPFC, MD, and HIPP but without a post hoc test to determine which specific suicide group was responsible for the significant differences [50]. Furthermore, there were no significant differences in microglial density between the suicidal and non-suicidal diagnostic groups (AD, MDD, and SCZ) and the control group in the DRN. However, the non-parametric Mann–Whitney U test showed increased microglial density in suicidal patients with depression compared to non-suicidal patients with depression [55].

An important microglial marker, allograft inflammatory factor 1 (AIF1; IBA1), its transcription in the dACC [56], and protein levels in the DLPFC, VLPFC [57] (activated phagocytes but not changed in IBA1), and blood vessels [56] of depressed suicides were increased in comparison with psychiatrically normal controls. Suicide patients with BD showed an increased IBA1 protein in the AmCC compared to non-suicide patients with BD, but this was not found in psychiatrically normal controls, nor in each group of suicide or non-suicide patients with SCZ [51]. However, there were no changes in *AIF1* mRNA levels in the ACC and DLPFC in depressed suicides [76].

QUIN protein, an endogenous modulator with agonistic properties on NMDA, which is produced by microglial cells, was significantly increased in the sACC and AmCC but not in the pACC in depressed suicidal patients compared to non-suicidal psychiatrically normal controls [52]. In contrast, suicidal patients with UD and MDD expressed decreased QUIN protein in the right hippocampal CA1 region compared with matched healthy controls [53]. Moreover, *PTPRC* transcript, also known as *CD45*, expressed in microglia, was significantly elevated in the depressed suicide group in the dACC. There were also significantly increased C-C motif chemokine ligand 2 (*CCL2*) transcripts in depressed suicide patients compared to psychiatrically normal controls [56]. Additionally, whole-transcriptome analysis has shown that depressed, suicide-associated genes are significantly enriched in microglia and microglial function [80,81].

#### 3.1.4. Oligodendrocytes

Oligodendrocytes are the myelinating cells of the CNS [27]. From our research, a total of nine studies assessed the alterations in the transcriptional, protein, and methylation levels of oligodendrocyte-related suicides (Figure 5 and Table 4). Chandley et al. investigated the expression of five antioxidant-related genes in oligodendrocytes captured from MDD suicide patients: superoxide dismutase-1 (*SOD1*), superoxide dismutase-2 (*SOD2*), glutathione peroxidase-1 (*GPX1*), catalase (*CAT*), and alkylglycerone phosphate synthase (*AGPS*). *AGPS* was significantly increased in the OCC and LC, whereas *SOD1*, *SOD2*, and *GPX1* were decreased in the OCC and LC, and *CAT* was increased in the OCC in suicides with MDD in comparison to psychiatrically normal controls [60]. Additionally, SOD2 (sp7823) protein was only confirmed in the aPFC and the MDD, BD, or drug dependence-related suicide groups but not in psychiatrically normal controls, which was interpreted as a compensatory upregulation of SOD2 protein in conditions of oxidative stress [66].

Tanti et al. and Honer et al., investigating the whole maturational stage of oligodendrocytes, determined that oligodendroglial abnormalities in the VMPFC white matter were seen in depressed suicide patients who experienced child abuse using specific markers. They observed decreased OLIG2 and MBP protein but increased achaete-scute homolog 1 (ASCL1; MASH1) protein of oligodendrocytes in these patients compared with psychiatrically normal controls [58,59]. However, no significant changes in the main constitutive proteins of myelin, including proteolipid protein 1 (PLP1), 2′,3′-cyclic nucleotide 3′ phosphodiesterase (CNP), myelin-associated glycoprotein (MAG), myelin oligodendrocyte glycoprotein (MOG), and myelin basic protein (MOBP), have been observed in suicidal patients with depression [59]. Their further RNA-sequencing study focused on gap junction coupling between astrocytes and oligodendrocytes in the ACC of depressed suicide patients and determined the changes in oligodendrocyte-related gene expression by quantitative real-time PCR, increased transcription of drebrin 1 (*DBN1*), and decreased transcriptions of gap junction protein beta 1 (*GJB1*), gap junction protein gamma (*GJC2*), and major tight junction proteins, such as occludin (*OCLN*) and zona occludens-1 and -2 (*TJP1/ZO-1* and *TJP2/ZO-2*), as well as an essential component of myelin interlamellar tight junctions, Claudin-11/OSP (*CLDN11*) [71]. Furthermore, genes encoding connexin-binding partners and regulators of GJ-mediated intercellular communication, such as caveolin-1 and -2 (*CAV1* and *CAV2*) and alpha/beta catenins (*CTNNA1* and *CTNNB1*), are decreased in depressed suicides, particularly in individuals with a history of child abuse [71]. Another study also confirmed increased transcription of *TJP1/ZO-1* and *CTNNA1* in the dACC of suicidal patients with depression compared to psychiatrically normal controls [56]. In addition, the transcription of an oligodendrocyte-specific RNA-binding protein, KH domain-containing RNA binding (*QKI*), which is important for cell development and myelination, was reduced in the OFC of suicide patients with depression compared to psychiatrically normal controls [82]. Recent RNA-Seq revealed the dysregulation of immature oligodendrocyte precursor cells in the DLPFC [83] and oligodendrocytes in the temporal pole (TP) in suicide patients with depression [61].

### 3.2. Alterations of Glial Cell Populations with Major Psychiatric Disorders

Although our review targeted previous literature that focused glial cells and their relationships with suicide, many studies used suicide decedents with psychiatric disorders in comparison with neuropsychiatric healthy age- and sex-matched controls as case–control studies. Each psychiatric disorder can be considered a confounding factor. Herein, we summarize 15 studies that directly determined changes in glial cells and their markers by comparing suicide and non-suicide patients with major psychiatric disorders, including AD, BD, MDD, and SCZ (Figure 6 and Table 5).

Among these studies, four literature compared glial density and the number of suicide decedents with psychiatric disorders in paired non-suicide patients, including alcohol dependence and early life adversity in patients with MDD [34,37,50,57]. Two showed significantly increased MHC-II protein microglia in BD suicides [54] and increased HLA-DR protein microgliosis in suicide patients with AD, BD, MDD, and SCZ (without a post hoc test) compared with non-suicide patients [50]. Three studies compared the suicidal patients with BD with non-suicidal patients with BD: IBA-1, MHC-II protein, and *P2RY12* transcript microglia, along with increased microglial marker *CD68* in the DLPFC, hippocampus, and AmCC [51,54,78]. In three studies of suicide patients with MDD, transcriptions of makers for astrocyte (*ALDH1L1*), microglia (*CX3CR1*, *CD68*, *ITGAM*, *HLA-DRA*, and *AIF1*), and oligodendrocytes protein (OLIG2, MOG, and PLP) were not changed in the ACC and DLPFC [76], and decreased transcripts in the PFC were significantly enriched in microglia [80] when compared with non-suicide patients with MDD. In addition, compared with suicide patients with MDD without a history of child abuse, those with a history of child abuse were confirmed to have oligodendroglial abnormalities, as previously described [59]. There were seven studies comparing suicide with non-suicide patients with SCZ, indicating suicide patients with SCZ displayed decreased transcriptions of astrocyte markers (*ALDH1L1* and *GLUL*) in the DLPFC and increased microglial markers (*CX3CR1*, *P2RY12*, and *TREM2*) in the ACC [65], and increased HLA-DR protein microglial density in the ACC, DLPFC, MD, and HIPP (without post hoc test) [49,50], while indicating decreased MBP protein in the anterior PFC (aPFC) [58] in comparison with non-suicide patients with SCZ.

In addition, two studies compared these major psychiatric disorders. For instance, Brisch et al. reported increased HLA-DR protein microglia in the DRN of suicide patients with BD, MDD, or SCZ compared to non-suicide patients (BD, MDD, or SCZ) but not significantly in each psychiatric disease group [55]. Furthermore, separating suicide means by violent and non-violent; those violent suicide patients with BD, MD, or SCZ exhibited increased transcription of microglial *P2RY12*, *P2RY13*, and *GPR34* [77].

## 4. Discussion

Glial cells have been reported to be involved in suicide, although the pathological mechanism of glial activity and the risk of suicidal behaviors remain unclear. Herein, we systematically review the literature on glial cell number, density, and cell type-specific markers in postmortem brains of suicides. In the literature, approximately 26% of the ACC and 30% of the DLPFC-related brain regions have been established. The densities and numbers of glial cells were changed by Nissl-glia, except for two studies that focused on the alterations of glial densities in the alcohol dependence and antidepressant treatment suicide group; no significant changes in glial cells were observed between suicides and their matched controls (Table 1). In addition, specific markers of glial cell types, such as GFAP (Golgi), CX3CR1, IBA1, and OLIG2, clarified each cell type, astrocytes, microglia, and oligodendrocytes, which displayed significant changes (Table 1), suggesting the limitation of staining techniques among the Nissl-, NeuN-, Golgi-, and glial-specific markers.

In our review, almost all suicide donors were patients with known psychiatric disorders, which is consistent with previous reports of the most common mental disorders in suicide [2]. However, considering the collection of sex- and age-matched non-suicide controls with psychiatric disorders are challenging, several studies directly compared the glial cells in psychiatric disorders diagnosed suicides with psychiatrically normal sudden-death controls. A previous review of postmortem studies indicated that glial cell abnormalities, including astrocytes, microglial cells, and oligodendrocytes, change with specific patterns in each psychiatric disease, such as BD, MDD, and SCZ, which may be confounding factors [84,85]. In our review, SCZ research did not find significant differences in HLA-DR-positive microglial density between patients with SCZ and psychiatrically normal controls (n = 16) in the DLPFC, ACC, HIPP, and MD related to sex or left vs. right hemisphere, whereas one patient with SCZ who committed suicide expressed higher microglial cell densities [49]. Their later study continued to show a reduction in microgliosis in suicides with SCZ/AD in comparison with non-suicide patients and psychiatrically normal controls by ANOVA, but without selecting a post hoc test there can be no confirmation of statistically significant differences in the suicide group with SCZ/MDD/AD in the DLPFC, ACC, and MD [50]. Therefore, despite the staining techniques, the densities and numbers of HLA-DR-positive microglia did not differ significantly between suicidal and non-suicidal patients with SCZ/MDD/AD or psychiatrically healthy controls.

Epidemiological studies have shown that clozapine reduces the risk of suicide in patients with SCZ [86], whereas clinical treatment-resistant depression has been established as a risk factor for suicide attempts and completed suicide [87]. In animal models, glial density is affected by antipsychotic medicines [88] and antidepressants [84]. In the current review, only one study showed significantly decreased Nissl-glial cell density in the anterior DG in suicide patients with antidepressant treatment in comparison with controls but not unmedicated suicide patients, which was insufficient even with relatively small sample sizes (n = 4) [41], suggesting that information on clinical medication histories, such as type, dosage, period, and frequency of antipsychotic medication, should be further discussed.

Our review clarifies the association between glia-related molecules and suicidal behaviors. Many astrocytic molecules (GFAP, Vimentin, ALDH1L1, SOX9, GLUL, SCL1A3, S100B, CX43, CX30, ICAM-1, CNR2, and GPR55) were decreased in the ACC, DLPFC, MD, PMC, PVC, CN, HIPP, LC, and Cb, while the transcription of astroglia-located components of the glutamate–glutamine cycle, EAAT1, EAAT2, and GLUL were decreased in the DLPFC of suicidal MDD patients (Table 2). Moreover, the different methylation levels of specific genes, hypomethylation of GRIK2 and NEBL, and hypermethylation of PVRL3 and ROPN1B were confirmed by the repressive histone modification of H3K9me3 in the prefrontal cortical astrocytes (Table 2). Additonally, many oligodendroglial molecules (OLIG2, Cx32, Cx47, OCLN, ZO-1, ZO-2, CLDN11, CAV1, CAV2, CTNNA1, CTNNB1, SOD1, and SOD2) showed decreased transcription and protein levels in suicide decedents (Table 4). In contrast, in microglia, almost all molecules were increased in the ACC, dACC, sACC, DLPFC, VLPFC, MD, AmCC, HIPP, DRN, and blood vessels, suggesting an increased activation of microgliosis in suicides (Table 3). As the literature on glial has been published so far, our finding is consistent with a previous review that astrocytic and oligodendroglial cells lost but activated microglia in major psychiatric disorders, including BD, MDD, and SCZ [89].

The neurophysiological and psychological interactions between suicidal behaviors and those changes in glial markers remain unclear. Indeed, suicide behaviors are a complex phenomenon that may be defined as a range of thinking about suicide, suicide threats, planning for suicide, attempting suicide, and suicide itself [90]. A recent meta-analysis suggested that the relationship between trait impulsivity and suicidal behavior is weak but occurs with extensive planning [91], suggesting long-term ideation before suicide completion could be determined by clinical features as well as biological risk. For instance, depressed patients exhibit agitation, marked irritability, weight loss, severe affective states, sleep disturbances, and self-injurious behaviors with suicidal intent [92]. Depressed patients who have attempted suicide also display increased levels of plasma interleukin (IL)-6 and TNF-α, as well as decreased IL-2 concentrations compared to non-suicidal depressed patients and psychiatrically normal controls [93]. Moreover, a meta-analysis showed lower plasma IL-2 levels in depressed suicidal patients in comparison with non-suicidal patients and healthy controls, while showing lower IL4 and higher TGF-β levels in suicidal patients in comparison with healthy controls [94]. Although immune activation has not been confirmed in the postmortem brain and cerebrospinal fluid of suicides, Snijders et al. reported the protein and mRNA levels of microglial activation markers, such as HLA-DRA, IL6, and IL1β, as well as the inflammatory responses to lipopolysaccharide and dexamethasone in patients with MDD, suggesting that microglia affect homeostatic functions that are essential for brain circuit development and maintenance and may be relevant for suicides [95]. We reported that the polymorphism of CX3CR1 is also related to the blood volume of the human brain [21]. Recent transcriptomic and meta-analyses have shown that altered transcription is associated with astrocytes and is dysregulated in suicidal behavior, mostly downregulated with pathways of protein translation, RNA metabolism [8], and GABAergic and glutamatergic neurotransmitters that may alter neuronal morphology [96]. Oligodendrocytes are crucial in the formation of the neurocircuitry for cognitive function, and their malfunction is reported to be associated with reduced myelin and oxidative stress deficiencies in depressed suicides [60]. Moreover, the gap junction proteins are reduced in glial cells of suicides [71], suggesting dysfunction of glial syncytium between astrocytes (A/A junctions), oligodendrocytes and astrocytes (O/A junctions), and neuron interactions.

Notably, each type of glial cell has specific functions but is also constituted together with neurons and endothelial cells to make up neurovascular unit components such as those of the BBB and mediate modulation of BBB permeability [15,97,98]. Increased BBB permeability may be a common pathological finding in MDD, BD, and SCZ [99]. Recently, Wu et al. reported increased levels of plasma intestinal fatty acid-binding protein (I-FABP), a biomarker of BBB permeability, in adolescent patients with MDD with moderate to severe anxiety [100]. Ventorp et al. also confirmed the increased hyaluronic acid in the cerebrospinal fluid of suicide attempters with increased BBB permeability [101]. As mentioned before, it seems that the density and number of astrocytes, microglia, and oligodendrocytes did not significantly change between suicide patients with MDD, BD, and SCZ and healthy subjects [34,36,37,38,39,49,50,85], considering that structural damage of the BBB, such as increased permeability, leads to the loss of BBB functional breakout and may result in suicidal thinking and behavior.

Additionally, enrichment analysis of the transcriptome revealed that differences in the BA46/9 were associated with suicide means (violent compared with non-violent), diagnosed psychiatric disorders, IQ, and suicide attempts [77]. Moreover, suicide is well known and associated with early adverse childhood experiences as well as adverse life events [38]. Proteomic and metabolomic studies of periphery blood indicated that inflammatory metabolism and imbalanced lipid transport are also associated with suicide behavior [96]. Together, suicide is a complex behavior that is rarely the outcome of a single factor, especially the molecular changes in glial cells. The above evidence indicates several possibilities that glial cell morphology, as well as the BBB, may be dysfunctional during suicide completion and need to be imaged using techniques such as positron emission tomography and single-photon emission tomography [79]. Furthermore, single-cell transcriptomic, proteomic, and metabolic analyses of postmortem brains in suicides could also improve the understanding of the molecular mechanism in the glial cells of suicide.

### Limitations

This review has several limitations. In terms of ethnicity, several studies have analyzed only European populations. Furthermore, brain regions, such as the left or right hemisphere and white or gray matter, were not discussed. The different methodologies and outcome measures identified with several patterns may affect the reliability and accuracy of the results. The sample size was small, which limited the comparison of each significant statistical analysis. Furthermore, it is challenging to combine the effect size by following statistical tests, such as meta-analysis, due to the need for the raw data in most studies in the current review. The effects of pharmacological and other medical therapies used in patients with psychiatric disorders on brain biology should be further discussed. Several concerns relate to sex differences, evaluation of the postmortem interval, and whether the patient died during the day or night, which may affect markers of glial activation after death [102].

## 5. Conclusions

Suicidal behavior is undoubtedly complex; some suicide risk factors are thought to contribute to the risk of suicidal behavior, including biological/individual, psychological, social, clinical/symptomatological, and environmental factors. However, whether glial cells contribute to suicidal behaviors remains unclear. In addition, no animal studies were included in this review. Although depression is a significant risk factor for suicide, suicidal behavior (e.g., sadness or suicidal ideation in depression) cannot be observed in animal models, and most stress exposure models mimic only mania- or depressive-like behaviors in the helplessness model. In our review, an animal model that determined that erb-B2 receptor tyrosine kinase 3 (ErbB3) expression was significantly decreased in the HIPP under chronic social defeat stress was similar to a postmortem study of depressed suicides [41]. However, evidence that does not represent animals clarifies the biological risk in glial cells for completing suicide, which is unable to untangle the etiological mechanisms underlying suicidality for prevention. Moreover, multi-level interventions should be the focus of further research because of their significant effects and synergistic potential. Beyond this consideration, modern suicide requires a better interpretation of suicide risk with a more careful assessment of suicide risk stratification and planning of clinical and treatment interventions, particularly among special populations. In conclusion, we investigated the relationship between alterations in glial cells and suicidal behaviors to provide new directions for suicide prevention.

## Figures and Tables

**Figure 1 ijms-25-05750-f001:**
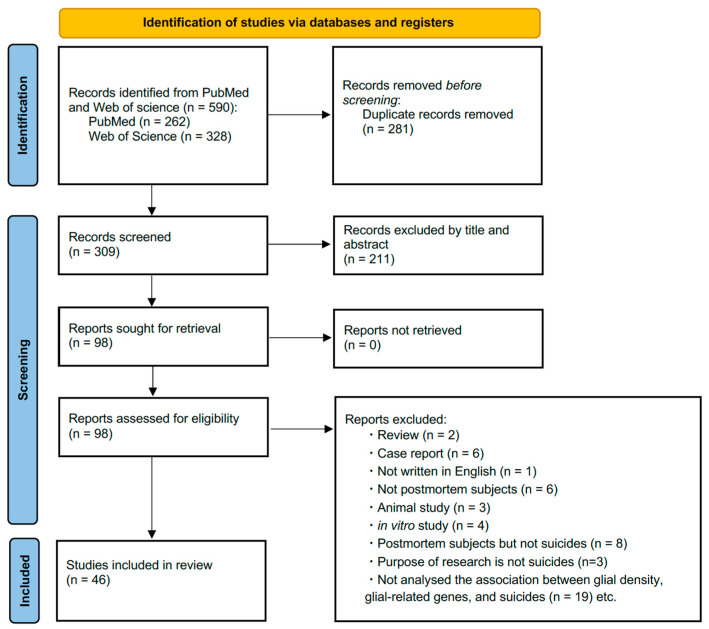
Study flow chart of the search to select studies that examined the association between glial cells and suicidal behavior.

**Figure 4 ijms-25-05750-f004:**
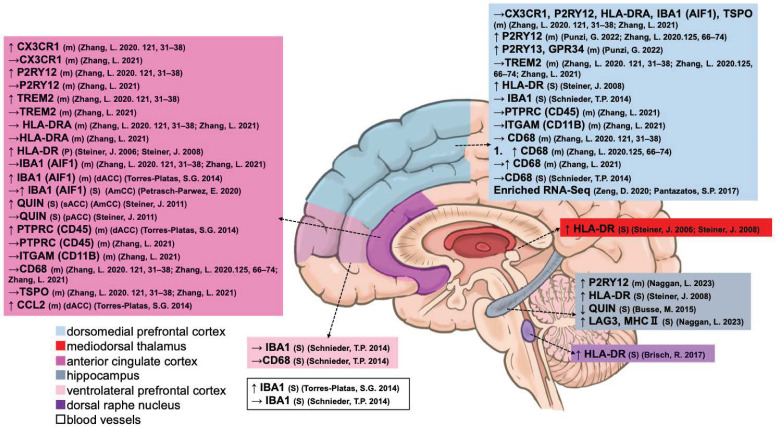
Postmortem evidence of microglial abnormalities in different brain regions. m, mRNA. P, protein. S, stain. D, DNA. ↑, increased. ↓, decreased. →, unchanged [50,51,52,56,57,65,76,77,78,80,81].

**Figure 5 ijms-25-05750-f005:**
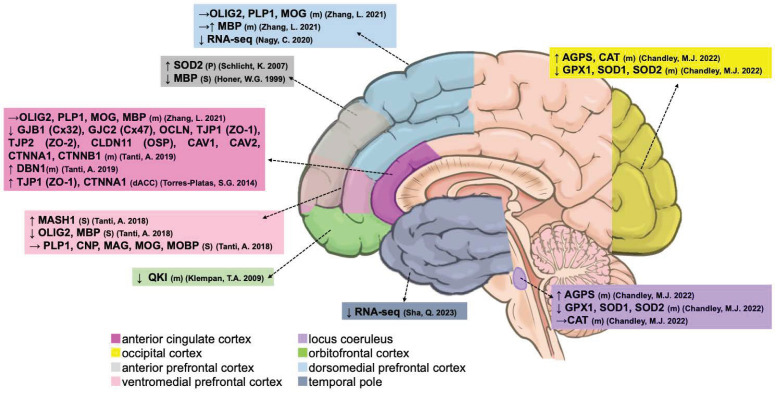
Postmortem evidence of oligodendroglial abnormalities in different brain regions. m, mRNA. P, protein. S, stain. D, DNA. ↑, increased. ↓, decreased. →, unchanged [56,58,59,60,61,66,71,76,82,83].

**Figure 6 ijms-25-05750-f006:**
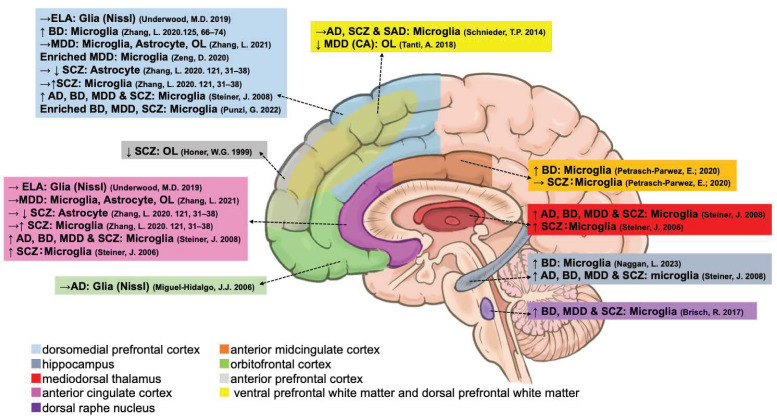
Morphological and molecular changes in glial cells in various psychiatric disorders. ↑, increased. ↓, decreased. →, unchanged [37,49,50,51,54,55,57,58,59,65,76,77,78,80].

**Table 3 ijms-25-05750-t003:** Changes in densities and numbers of microglia in suicides.

Microglia	Type	ACC	dACC	sACC	pACC	DLPFC	VLPFC	MD	AmCC	HIPP	DRN	Blood Vessels
*CX3CR1*	mRNA	↑ [65] → [76]	—	—	—	→ [65,76]	—	—	—	—	—	—
*P2RY12*	mRNA	↑ [65] → [76]	—	—	—	↑ [77,78] → [65,76]	—	—	—	↑ [54]	—	—
*P2RY13*	mRNA	—	—	—	—	↑ [77]	—	—	—	—	—	—
*GPR34*	mRNA	—	—	—	—	↑ [77]	—	—	—	—	—	—
*TREM2*	mRNA	↑ [65] → [76]	—	—	—	→ [65,76,78]	—	—	—	—	—	—
*HLA-DRA*	mRNA	→ [65,76]	—	—	—	→ [65,76]	—	—	—	—	—	—
HLA-DR	Stain	↑ [49] ^#^, [50] ^$^	—	—	—	↑ [50] ^$^	—	↑ [49] ^#^, [50] ^$^	—	↑ [50] ^$^	↑ [55]	—
*IBA1 (AIF1)*	mRNA	→ [65,76]	↑ [56]	—	—	→ [65,76]	—	—	—	—	—	—
IBA1	Stain	—	—	—	—	→ [57]	→ [57]	—	→↑ [51]	—	—	↑ [56] → [57]
QUIN	Stain	—	—	↑ [52]	→ [52]	—	—	—	↑ [52]	↓ [53]	—	—
*PTPRC* *(CD45)*	mRNA	→ [76]	↑ [56]	—	—	→ [76]	—	—	—	—	—	—
*ITGAM (CD11B)*	mRNA	→ [76]	—	—	—	→ [76]	—	—	—	—	—	—
*CD68*	mRNA	→ [65,76,78]	—	—	—	→ [65] ↑ [78] →↑ [76]	—	—	—	—	—	—
CD68	Stain	—	—	—	—	→ [57]	→ [57]	—	—	—	—	—
*TSPO*	mRNA	→ [65,76]	—	—	—	→ [65,76]	—	—	—	—	—	—
LAG3	Stain	—	—	—	—	—	—	—	—	↑ [54]	—	—
MHC II	Stain	—	—	—	—	—	—	—	—	↑ [54]	—	—
*CCL2*	mRNA	—	↑ [56]	—	—	—	—	—	—	—	—	—
RNA-Seq	mRNA	—	—	—	—	Enriched [80]	—	—	—	—	—	—
RNA-Seq	mRNA	—	—	—	—	Enriched [81]	—	—	—	—	—	—

CX3CR1, C-X3-C motif chemokine receptor 1. P2RY12, purinergic receptor P2Y12. P2RY13, pu-rinergic receptor P2Y13. GPR34, G protein-coupled receptor 34. HLA-DR, human leukocyte anti-gen-DR isotype. HLA-DRA, human leukocyte antigen-DR Alpha. IBA1, allograft inflammatory factor 1 (AIF-1). QUIN, quinolinic acid. TREM2, triggering receptor expressed on myeloid cells 2. CD45, protein tyrosine phosphatase receptor type C (PTPRC). CD11B, integrin subunit alpha M (ITGAM). CD68, CD68 molecular. TSPO, translocator protein. CCL2, C-C motif chemokine 2. LAG3, lymphocyte activation gene 3. MHC II, major histocompatibility complex class II. ACC, anterior cingulate cortex. dACC, dorsal anterior cingulate cortex. sACC, subgenual anterior cin-gulate cortex. pACC, perigenual anterior cingulate cortex. DLPFC, dorsolateral prefrontal cortex. VLPFC, ventrolateral prefrontal cortex. MD, mediodorsal nucleus. AmCC, anterior midcingulate cortex. HIPP, hippocampus. DRN, dorsal raphe nucleus. ^#^, without statistical analysis. ^$^, without post hoc statistical significance. ↑, increased; ↓, decreased; →, unchanged; — not applicable.

**Table 4 ijms-25-05750-t004:** Changes in densities and numbers of oligodendroglial cells in suicides.

Oligodendrocyte	Type	ACC	dACC	aPFC	DLPFC	VMPFC	LC	OCC	OFC	TP
*AGPS*	mRNA	—	—	—	—	—	↑ [60]	↑ [60]	—	—
*SOD1*	mRNA	—	—	—	—	—	↓ [60]	↓ [60]	—	—
*SOD2*	mRNA	—	—	—	—	—	↓ [60]	↓ [60]	—	—
SOD2	Protein	—	—	↑ [66]	—	—	—	—	—	—
*GPX1*	mRNA	—	—	—	—	—	↓ [60]	↓ [60]	—	—
*CAT*	mRNA	—	—	—	—	—	→ [60]	↑ [60]	—	—
*OLIG2*	mRNA	→ [76]	—	—	→ [76]	—	—	—	—	—
OLIG2	Stain	—	—	—	—	↓ [59]	—	—	—	—
*PLP1*	mRNA	→ [76]	—	—	→ [76]	—	—	—	—	—
PLP1	Stain	—	—	—	—	→ [59]	—	—	—	—
CNP	Stain	—	—	—	—	→ [59]	—	—	—	—
MAG	Stain	—	—	—	—	→ [59]	—	—	—	—
MASH1	Stain	—	—	—	—	↑ [59]	—	—	—	—
*MOG*	mRNA	→ [76]	—	—	→ [76]	—	—	—	—	—
MOG	Stain	—	—	—	—	→ [59]	—	—	—	—
MOBP	Stain	—	—	—	—	→ [59]	—	—	—	—
*MBP*	mRNA	→ [76]	—	—	→↑ [76]	—	—	—	—	
MBP	Stain	—	—	↓ [58]	—	↓ [59]	—	—	—	—
*GJB1 (Cx32)*	mRNA	↓ [71]	—	—	—	—	—	—	—	—
*GJC2 (Cx47)*	mRNA	↓ [71]	—	—	—	—	—	—	—	—
*OCLN*	mRNA	↓ [71]	—	—	—	—	—	—	—	—
*TJP1 (ZO-1)*	mRNA	↓ [71]	↑ [56]	—	—	—	—	—	—	—
*TJP2 (ZO-2)*	mRNA	↓ [71]		—	—	—	—	—	—	—
*CLDN11 (OSP)*	mRNA	↓ [71]	—	—	—	—	—	—	—	—
*CAV1*	mRNA	↓ [71]	—	—	—	—	—	—	—	—
*CAV2*	mRNA	↓ [71]	—	—	—	—	—	—	—	—
*CTNNA1*	mRNA	↓ [71]	↑ [56]	—	—	—	—	—	—	—
*CTNNB1*	mRNA	↓ [71]	—	—	—	—	—	—	—	—
*DBN1*	mRNA	↑ [71]	—	—	—	—	—	—	—	—
*QKI*	mRNA	—	—	—	—	—	—	—	↓ [82]	—
RNA-Seq	mRNA	—	—	—	↓ [83]	—	—	—	—	↓ [61]

SOD1, superoxide dismutase-1. SOD2, superoxide dismutase-2. AGPS, alkylglycerone phosphate synthase. CAT, catalase. GPX1, glutathione peroxidase-1. GJB1, gap junction protein beta 1. GJC2, gap junction protein gamma 2. OLIG2, oligodendrocyte transcription factor 2. PLP1, proteolipid protein 1. MASH1, Achaete-scute homolog 1 (ASCL1). CNP, 2′,3′-cyclic nucleotide 3′ phos-phodiesterase. MAG, myelin-associated glycoprotein. MOG, myelin oligodendrocyte glycopro-tein. MOBP, myelin basic protein. MBP, myelin basic protein. OCLN, occludin. TJP1 (ZO-1), tight junction protein 1. TJP2 (ZO-2), tight junction protein 2. CLDN11, Claudin-11. CAV1, caveolin-1. CAV2, caveolin-2. CTNNA1, Catenin Alpha 1. CTNNB1, Catenin Beta 1. DBN1, Drebrin 1. QKI, KH domain containing RNA binding. ACC, anterior cingulate cortex. dACC, dorsal anterior cingulate cortex. aPFC, anterior prefrontal cortex. DLPFC, dorsolateral prefrontal cortex. VMPFC, ventromedial prefrontal cortex. LC, locus coeruleus. OCC, occipital cortex. OFC, orbitofrontal cortex. TP, temporal pole. ↑, increased; ↓, decreased; →, unchanged; — not applicable.

**Table 5 ijms-25-05750-t005:** Changes in glial cells in suicides with psychiatric disorders.

Diagnoses	Cell Type	Changes	Region
Ad	Glia (Nissl)	→ [34]	OFC
ELA	Glia (Nissl)	→ [37]	ACC, DLPFC
BD	Microglia	↑ [54]	HIPP
BD	Microglia	↑ [78]	DLPFC
BD	Microglia	↑ [51]	AmCC
MDD	Astrocyte	→ [76]	ACC, DLPFC
MDD	Microglia	→ [76]	ACC, DLPFC
MDD	OL	→ [76]	ACC, DLPFC
MDD	Microglia	enriched [80]	DLPFC
SCZ	Astrocyte	→↓ [65]	ACC, DLPFC
SCZ	Microglia	→↑ [65]	ACC, DLPFC
AD, BD, MDD, and SCZ	Microglia	↑ [50] ^$^	ACC, DLPFC, HIPP, MD
SCZ	Microglia	↑ [49] ^#^	ACC, MD
AD, SCZ, and SAD	Microglia	→ [57]	VD
SCZ	Microglia	→ [51]	AmCC
SCZ	OL	↓ [58]	aPFC
BD MDD SCZ	Microglia	↑ [55]	DRN
MDD (CA)	OL	↓ [59]	VD
BD MDD SCZ	Microglia	enriched [77]	DLPFC

Ad, alcohol dependence; AD, affective (mood) disorders; BD, bipolar disorder; ACC, anterior cingulate cortex; AmCC, anterior midcingulate cortex; aPFC, anterior prefrontal cortex; CA, child abuse; DLPFC, dorsomedial prefrontal cortex; DRN, dorsal raphe nucleus; ELA, early-life adversity/child abuse; HIPP, hippocampus; MD, mediodorsal thalamus; MDD, major depressive disorder; OFC, orbitofrontal cortex; OL, oligodendrocyte; SAD, schizoaffective disorder; SCZ, schizophrenia; VD, ventral prefrontal white matter and dorsal prefrontal white matter. ^$^, without post hoc statistical significance. ^#^, without statistical analysis. ↑, increased; ↓, decreased; →, unchanged.

## Data Availability

Not applicable.

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
