# Peer review of "Glial Markers of Suicidal Behavior in the Human Brain—A Systematic Review of Postmortem Studies"

_ijms, 2024, doi:10.3390/ijms25115750_

Round 1

Reviewer 1 Report

Comments and Suggestions for Authors This manuscript reviews relevant studies to explore and synthesize evidence of the relationship between glial dysfunction and suicidal behavior to Understanding the neurobiology of suicide has some value. There are also questions that the authors need to consider.

1. It is suggested that the authors further elucidate and analyze the neurophysiological and psychological interactions between suicidal behavior and glial markers, as well as the mechanism of their interaction.
2. In the discussion of various glial cell changes, some sections of the manuscript did not use appropriate statistical methods to determine the significance of the results. For example, in relation to suicide versus non-suicide in patients with SCZ, changes in glial cell density were mentioned, but no follow-up statistical tests were performed to confirm the significance of these differences. A complete statistical analysis of all significance tests is recommended and specific p-values and effect sizes are provided in the manuscript.

Author Response

Thank you for your valuable comments. We have uploaded the revised manuscript with changes underlined and marked in red.

Please find below our point-by-point response to individual comments.

Reviewer 1

This manuscript reviews relevant studies to explore and synthesize evidence of the relationship between glial dysfunction and suicidal behavior to Understanding the neurobiology of suicide has some value. There are also questions that the authors need to consider.

1-1. It is suggested that the authors further elucidate and analyze the neurophysiological and psychological interactions between suicidal behavior and glial markers, as well as the mechanism of their interaction.

Response:
We have discussed the interactions between suicidal behaviors and markers of glial cells in the “Discussion” as described below.

Page 14, Line 487 – Page 15, Line 542:

The neurophysiological and psychological interactions between suicidal behaviors and those changes in glial markers remain unclear. Indeed, suicide behaviors are a complex phenomenon that may be defined as a range of thinking about suicide, suicide threats, planning for suicide, attempting suicide, and suicide itself [90]. A recent meta-analysis suggested that the relationship between trait impulsivity and suicidal behavior is weak but occurs with extensive planning [91], suggesting long-term ideation before suicide completion could be determined by clinical features as well as biological risk. For instance, depressed patients exhibit agitation, marked irritability, weight loss, severe affective states, sleep disturbances, and self-injurious behaviors with suicidal intent [92]. Depressed patients who have attempted suicide also display increased levels of plasma interleukin (IL)-6 and TNF-α, as well as decreased IL-2 concentrations compared to non-suicidal de-pressed patients and psychiatrically normal controls [93]. Moreover, a meta-analysis showed lower plasma IL-2 levels in depressed suicidal patients in comparison with non-suicidal patients and healthy controls, while lower IL4 and higher TGF-β levels in suicidal patients in comparison with healthy controls [94]. Although immune activation has not been confirmed in postmortem brain and cerebrospinal fluid of suicides, Snijders et al. reported the protein and mRNA levels of microglial activation markers, such as HLA-DRA, IL6, and IL1β, as well as the inflammatory responses to lipopolysaccharide and dexamethasone in patients with MDD, suggesting microglia affect homeostatic functions that essential for brain circuit development and maintenance, may be relevant for suicides [95]. We reported that he polymorphism of CX3CR1 is also related to the blood volume of the human brain [21]. Recent transcriptomic and meta-analyses have shown that altered transcription is associated with astrocytes and is dysregulated in suicidal behavior, mostly downregulated with pathways of protein translation, RNA metabolism [96], GABAergic and glutamatergic neurotransmitters that may alter neuronal mor-phology [97]. Oligodendrocytes are crucial in the formation of the neurocircuitry for cognitive function, and their malfunction is reported to be associated with reduced myelin and oxidative stress deficiencies in depressed suicides [60]. Moreover, the gap junction proteins are reduced in glial cells of suicides [71], suggesting dysfunction of glial syn-cytium between astrocytes (A/A junctions), oligodendrocytes and astrocytes (O/A junc-tions), and neuron interactions.

            Notably, each type of glial cell has specific functions but is also constituted together with neurons and endothelial cells to make up neurovascular unit components such as those of the BBB and mediate modulation of BBB permeability [15, 98]. Increased BBB permeability may be a common pathological finding in MDD, BD, and SCZ [99]. Recently, Wu et al. reported increased levels of plasma intestinal fatty acid-binding protein (I-FABP), a biomarker of BBB permeability, in adolescent patients with MDD with moderate to severe anxiety [100]. Ventorp et al. also confirmed the increased hyaluronic acid in the cerebrospinal fluid of suicide attempters with increased BBB permeability [101]. As mentioned before, it seems that the density and number of astrocytes and oligoden-drocytes did not significantly change between suicide patients with MDD, MD, and SCZ and healthy subjects [85], considering that structural damage of the BBB, such as increased permeability, leads to the loss of BBB functional breakout and may result in suicidal behavior.

            Additionally, enrichment analysis of the transcriptome revealed that differences in the DLPFC were associated with suicide means (violent compared with non-violent), diagnosed psychiatric disorders, IQ, and suicide attempts [77, 80]. Moreover, suicide is well known and associated with early adverse childhood experiences as well as adverse life events [38]. Proteomic and metabolomic studies of periphery blood indicated that inflammatory metabolism and imbalanced lipid transport are also associated with suicide behavior [97]. Together, suicide is a complex behavior that is rarely the outcome of a single factor, especially the molecular changes in glial cells. The above evidence indicates several possibilities that glial cell morphology, as well as the BBB, may be dysfunctional during suicide completion, and need to be imaged using techniques such as positron emission tomography and single photon emission tomography. Furthermore, single-cell transcriptomic, proteomic, and metabolic analyses of postmortem brains in suicides could also improve the understanding of the molecular mechanism in the glial cells of suicide.

1-2. In the discussion of various glial cell changes, some sections of the manuscript did not use appropriate statistical methods to determine the significance of the results. For example, in relation to suicide versus non-suicide in patients with SCZ, changes in glial cell density were mentioned, but no follow-up statistical tests were performed to confirm the significance of these differences. A complete statistical analysis of all significance tests is recommended and specific p-values and effect sizes are provided in the manuscript.

Response:

We recognized that versatile and powerful statistical tests will give greater statistical power to provide accurate results between glial changes and suicidal behaviors for our review. However, many of the studies in our review did not provide raw data (mean, standard deviation), an inability to perform further statistical tests. We have addressed this issue in the section of Limitation.

Page 15, Line 549-552:

Furthermore, it is challenging to combine the effect size by following statistical tests, such as meta-analysis, due to the need for the raw data in most studies in the current review.

Reviewer 2 Report

Comments and Suggestions for Authors

This manuscript analyses of the interesting and quite rare issue of deep analysis of post-mortem brains after suicidal death.

The Methodology of this systematic review is well presented and described in details.

The main objection to this manuscript is not using the gene symbol and formatting conventions. Ale the gene names should be written with italic, similarly their mRNA and cDNA names and only protein should be written as plain upper-case letters.

In analyses where protein is studied, it is worth to mentioned that this is XXX protein.

The caption for Figure 2 could be more differential than for Figure 6

However, I am not authorized to assess language quality, but I have read some sentences a too general, like for example, the first sentence in 3.4. Oligodendrocytes, the caption of the Tables and others.

The Limitations section should contain additionally the reservation considering different methodology used in all those reviewed studies, making difficult the reliable comparisons.

Author Response

Thank you for your valuable comments. We have uploaded the revised manuscript with changes underlined and marked in red.

Reviewer 2

This manuscript analyses of the interesting and quite rare issue of deep analysis of post-mortem brains after suicidal death.

The Methodology of this systematic review is well presented and described in details.

2-1. The main objection to this manuscript is not using the gene symbol and formatting conventions. Ale the gene names should be written with italic, similarly their mRNA and cDNA names and only protein should be written as plain upper-case letters.

Response:

We have carefully revised the manuscript to adhere to the proper gene and protein notation conventions.

2-2. In analyses where protein is studied, it is worth to mentioned that this is XXX protein.

Response:

We have revised our manuscript, including the changes to gene and protein symbols mentioned above.

2.3 The caption for Figure 2 could be more differential than for Figure 6

Response:

We have revised the captions accordingly to better reflect each figure's unique aspects and findings, as shown below.

Page 7, Line 220:

Figure 2. Postmortem evidence of glial abnormalities in different brain regions.

Page 12, Line 425-426:

Figure 6. Morphological and molecular changes of glial cells in various psychiatric disorders.

2.4 However, I am not authorized to assess language quality, but I have read some sentences a too general, like for example, the first sentence in 3.4. Oligodendrocytes, the caption of the Tables and others.

Response:

We have revised these sections to provide more specific and detailed information that better supports our findings, including the first sentence in section 3.4., as described below.

Page 7, Line 223-224:

Astrocytes are the most functionally diverse glial cells in the CNS, including modulation of BBB and their permeability, and provision of nutrients to the nervous tissue [15].

Page 8, Line 283-285:

As the primary immune cell of the CNS [9], sixteen papers reported on the alterations in transcription, protein content, and methylation of microglia-related suicides (Figure 4 and Table 3).

Page 10, Line 342-343:

Oligodendrocytes are the myelinating cells of the CNS [27]. From our research, a total of nine studies assessed the alterations in the transcriptional, protein, and methylation levels of oligodendrocyte-related suicides (Figure 5 and Table 4).

2.5 The Limitations section should contain additionally the reservation considering different methodology used in all those reviewed studies, making difficult the reliable comparisons.

Response:

We recognize the importance of addressing the variability in methodology across the reviewed studies and the challenges this poses for making reliable comparisons. We have revised the "limitations" section as described below.

Page 15, Line 546-547:

The different methodologies and outcome measures identified with several patterns may affect the reliability and accuracy of results.